

# Time evolution of temperature profiles retrieved from 13 years of IASI data using an artificial neural network

Marie Bouillon[1], Sarah Safieddine[1], Simon Whitburn[2], Lieven Clarisse[2], Filipe Aires[3], Victor Pellet[3], Olivier Lezeaux[4], Noëlle A. Scott[5], Marie Doutriaux-Boucher[6], Cathy Clerbaux[1,2]

[1]LATMOS/IPSL, Sorbonne Université/UVSQ/CNRS, Paris, France
[2]Spectroscopy, Quantum Chemistry and Atmospheric Remote Sensing (SQUARES), Université Livre de Bruxelles (ULB), Brussels, Belgium
[3]LERMA, CNRS, Paris, France
[4]SPASCIA, Ramonville-Saint-Agne, France
[5]Laboratoire de Météorologie Dynamique, IPSL/CNRS/Ecole Polytechnique/Université Paris-Saclay, Palaiseau, France
[6]European Organisation for the Exploitation of Meteorological Satellites, Darmstadt,  Germany

*Correspondence to*: Marie Bouillon (marie.bouillon@latmos.ipsl.fr)

**Abstract.** The three IASI instruments, launched in 2006, 2012, and 2018, are key instruments to weather forecasting, and

most meteorological centers assimilate IASI nadir radiance data into atmospheric models to feed their forecasts. The EUropean organisation for the exploitation of METeorological SATellites (EUMETSAT) recently released a reprocessed homogeneous radiance record for the whole IASI observation period, from which thirteen years (2008-2020) of temperature profiles can be obtained. In this work, atmospheric temperatures at different altitudes are retrieved from IASI radiances measured in the carbon dioxide absorption bands (654-800 cm$^{-1}$ and 2250-2400 cm$^{-1}$) by selecting the channels that are the

most sensitive to the temperature at different altitudes. We rely on an Artificial Neural Network (ANN) to retrieve atmospheric temperatures from a selected set of IASI radiances. We trained the ANN with IASI radiances as input and the European Centre for Medium-range Weather Forecasts (ECMWF) reanalysis version 5 (ERA5) as output. The retrieved temperatures were validated with ERA5, with in-situ radiosonde temperatures from the Analysed RadioSoundings Archive (ARSA) network and with EUMETSAT temperatures retrieved from IASI radiances using a different method. Between 750

and 7 hPa, where IASI is most sensitive to temperature, a good agreement is observed between the three datasets: the differences between IASI on one hand, and ERA5, ARSA or EUMETSAT on the other hand are usually less than 0.5 K at these altitudes. At 2 hPa, as the IASI sensitivity decreases, we found differences up to 2 K between IASI and the three validation datasets. We then computed atmospheric temperature linear trends from atmospheric temperatures between 750 and 2 hPa. We found that in the past thirteen years, there is a general warming trend of the troposphere, that is more

important at the poles than at the equator (0.7 K/decade at the equator, 1 K/decade at the North Pole). The stratosphere is globally cooling on average, except at the South Pole as a result of the ozone layer recovery. The cooling is most pronounced in the equatorial upper stratosphere (-1 K/decade). This work shows that ANN can be a powerful and simple tool to retrieve



IASI temperatures at different altitudes in the upper troposphere and in the stratosphere, allowing us to construct a homogeneous and consistent temperature data record adapted to trend analyis.

## 1 Introduction

Atmospheric temperatures are a key component of Earth's climate. In the past few decades, a warming of the troposphere due to the increase of greenhouse gas concentrations (Tett et al., 2002; Santer et al., 2017; Susskind et al., 2019; Shanbaum et al., 2014) and a cooling of the stratosphere have been observed (Randel et al., 2016; Maycock et al., 2018). Stratospheric temperatures are impacted by both anthropogenic forcing (e.g. greenhouse gas emissions, ozone depletion) and natural forcing (e.g. volcanic eruptions, solar cycle) (Aquila et al., 2016). The study of stratospheric temperatures and their long-term evolution is therefore critical to understand the roles of these different forcings on the evolution of climate in the stratosphere, but also in the troposphere.

Long term atmospheric temperature records are obtained from in situ measurements (lidars and radio soundings). These observations are generally of excellent quality, however, they are sparse and unevenly distributed around the globe. More recently, satellite-derived temperatures have become a key component for climate change monitoring (Li et al., 2011; Yang et al., 2013). Satellite observations have a better spatial coverage but the construction of a long temperature record from these observations usually requires merging several different instruments, and corrections and adjustments between the observations are needed to obtain a homogeneous dataset (Zou et al., 2014; Seidel et al., 2016).

In 2006, the first Infrared Atmospheric Sounding Interferometer (IASI) was launched on the Metop satellite. IASI measures radiance spectra from which surface and atmospheric temperatures (Hilton et al., 2012; Safieddine et al., 2020a) and trace gas concentrations can be retrieved (Clerbaux et al., 2009; Clarisse et al., 2011). A second and a third instruments were launched in 2012 and 2018, and the comparison between the three instruments have shown excellent agreement (Boynard et al., 2018; EUMETSAT, 2013a). Since IASI is planned to fly for at least 18 years, with the three instruments built at the same time and flying in constellation, continuity and stability are insured, and the potential of constructing a long-term climate data record at a range of altitudes is becoming evident.

IASI radiance spectra and derived atmospheric temperature profiles are routinely processed by the European organisation for the exploitation of METeorological SATellites (EUMETSAT). Over the past 13 years, EUMETSAT has performed several updates on the processing of both radiances and temperatures, making the time series non-homogeneous. The impacts of these updates have been evidenced in several studies (George et al., 2015; Van Damme et al., 2017; Parracho et al., 2021) and quantified in Bouillon et al. (2020). For temperatures, the "jumps" in the time series due to these updates make them unfit for the computation of trends. In 2018, EUMETSAT reprocessed Metop-A radiance dataset



(doi:10.15770/EUM_SEC_CLM_0014) , providing a new radiance dataset over time. After 2018, the radiances are stable and consistent with those reprocessed (Bouillon et al., 2020).

In this work, we present a new atmospheric temperature product derived from the homogeneous IASI radiance dataset, using an Artificial Neural Network (ANN) technique, in order to derive a homogeneous temperature data record. In Sect. 2, we
explain the method used to compute the temperatures, both the selection of IASI channels and the training of the ANN. In Sect. 3, we compare the outputs of the neural network with both the latest ECMWF reanalysis (ERA5) and ARSA radiosondes temperatures to validate the new data. In Sect. 4 we compute atmospheric temperature trends for the past 13 years. Conclusions are listed in Sect. 5.

## 2 Methods

### 2.1 The IASI instrument channel selection

Each of the three IASI instruments are mounted on-board the Metop platform flying on a polar orbit at an altitude of 817 km. The IASI swath contains 30 fields of view with 4 pixels in each field of view. This observation mode allows each IASI instrument to observe the entire Earth twice a day, between 9:15 and 9:45 AM and PM local time. IASI measures the radiation of the Earth-atmosphere system in the thermal infrared in 8461 channels between 645 and 2760 cm$^{-1}$ (resolution of
0.25 cm$^{-1}$, 0.5 cm$^{-1}$ apodized; Clerbaux et al., 2009).

Using the 8461 channels of IASI raises practical issues for storage and computation power, as retrieval and assimilation algorithms can hardly handle such a large amount of information. A channel selection is usually needed when dealing with IASI (Rabier et al., 2002; Collard, 2007; Pellet and Aires, 2018). To retrieve atmospheric temperatures, we select IASI
channels that are most sensitive to the temperature profile. Most of the channels selected are located in the carbon dioxide ($CO_2$) absorption band, because the radiances observed in these channels is mostly impacted by atmospheric temperatures, rather than $CO_2$ concentrations (Chédin et al., 2003; Collard, 2007). The selection is obtained using the Entropy Reduction (ER) method (Rodgers, 2000). The entropy describes the probabilities of all the possible states and it is maximal when all the states have an equal probability. Selecting the channels that reduce the most the entropy means selecting the channels that
bring the most information about the state. ER is computed using:

$$ER = \frac{1}{2}\log_2(\mathbf{BA^{-1}}) \qquad (1)$$

where $\mathbf{B}$ is the a priori covariance matrix and $\mathbf{A}$ is the retrieval covariance matrix, described in the following Eq. 2:

$$\mathbf{A} = (\mathbf{B^{-1}} + \mathbf{H^T R^{-1} H})^{-1} , \qquad (2)$$



where **H** is the matrix of the temperature Jacobians (the sensitivity of the IASI brightness temperature to the temperature),
and **R** is the instrumental noise plus the radiative transfer error.

With **A**, it is possible to compute the entropy reduction of each channel as such:

$$\delta ER = \frac{1}{2}\log_2\left(1 + \mathbf{h'}^T \mathbf{A}_{i-1}\mathbf{h'}\right),$$    (3)

with **h'** being the Jacobian of the considered channel normalized by the noise ($\mathbf{H'} = \mathbf{R}^{-1/2}\mathbf{H}$). For the selection of the first
channel, we set $\mathbf{A}_0 = \mathbf{B}$. With this, we selected the channel with the largest entropy reduction, and the theoretical covariance
matrix is updated as follows:

$$\mathbf{A}_i = \mathbf{A}_{i-1} - \frac{(\mathbf{A}_{i-1}\mathbf{h'})(\mathbf{A}_{i-1}\mathbf{h'})^T}{1 + (\mathbf{A}_{i-1}\mathbf{h'})^T\mathbf{h'}}.$$    (4)

We then repeat this process until the chosen number of channels have been selected or until the entropy has been reduced
enough. This method has been used to retrieve skin temperature from IASI (Safieddine et al., 2020a), and we apply it here
for atmospheric temperature profiles.

In order to take into account the effect of the different parameters affecting the selection, two experiments were made: in the
first, we considered 200 channels. The channels were selected while taking into account the perturbation of the radiances due
to water vapour ($H_2O$) and ozone ($O_3$) variations because some channels are sensitive to temperature, $H_2O$ and $O_3$ (Pellet and
Aires, 2018). The uncertainty on the state of unretrieved species (i.e $H_2O$ and $O_3$) impacts the potential retrieval of the
temperature using these channels. This perturbation is then computed with $\mathbf{H}_x \times \mathbf{B}_x \times \mathbf{H}_x^T$ (with $x$ being the variable
considered: $H_2O$ or $O_3$) and is added to the instrumental noise and radiative transfer error, so **A** becomes:

$$\mathbf{A} = \left\{\mathbf{B}_{temp}^{-1} + \mathbf{H}_{temp}^T \left(\mathbf{R} + \mathbf{H}_{H2O}\mathbf{B}_{H2O}\mathbf{H}_{H2O}^T + \mathbf{H}_{O3}\mathbf{B}_{O3}\mathbf{H}_{O3}^T\right)^{-1}\mathbf{H}_{temp}\right\}^{-1}.$$    (5)

The second experiment consists of 100 channels. Before the ER method is applied to all the channels, the channels for which
the variability of atmospheric gases ($H_2O$, $O_3$, CO and $CH_4$) and emissivity has an impact higher than 30% of the
instrumental noise are removed.


In the first selection, the Jacobians were computed with the Optimal Spectral Sampling (OSS) radiative transfer model
(Moncet et al., 2015). In the second selection, the model used was Radiative Transfer for TOVS (RTTOV, Saunders et al.,
2018).

For each of the two selection methods, the number of channels selected is increased until adding more new channels does not



significantly improves the results.

The goal of using these two sets of experiments is to choose from these two the best and most sensitive channels to different atmospheric temperatures while taking into account the different atmospheric perturbations and errors that might affect the
selection. On its own, each experiment was tested (not shown here), and the best result was achieved when combining them both.

For the computation of temperatures, we used a mix of the two experiments, consisting of 231 channels (with 69 channels in common between the two). Figure 1 shows the selected channels on a typical IASI spectrum. Most of the channels are in the
$v_2$ carbon dioxide ($CO_2$) absorption band between 645 and 800 cm$^{-1}$ while few (14 channels) are at 2200 cm$^{-1}$ in $v_1$ the $N_2O$ absorption band. The full list of the channels is provided in Supplementary material (Table S1).

## 2.2 Artificial neural network

We trained a two-layer artificial neural network (ANN) to estimate atmospheric temperature profiles. This method has been used for instance in Aires et al., 2002, using IASI simulated radiances before its launch.
450 000 IASI observations are used to train the ANN. These observations (3000 scenes per month) were selected randomly around the globe between January 2008 and December 2020. This training dataset is composed of day and night, clear and cloudy sky observations mixed together. The input data consists of the pseudo-normalised radiances (multiplied by $10^4$, so that their order of magnitude is not too small compared to temperatures) in the selected channels, as well as the scan angle of
the observation. A global monthly $CO_2$ was also added to take into account the $CO_2$ variations that impact the radiance values measured in the selected channels. The $CO_2$ monthly values come from the NOAA-ESRL global monitoring dataset (https://gml.noaa.gov/ccgg/trends/). As the expected output of the training of the ANN, we use the ERA5 temperatures interpolated to the latitudes, longitudes, and time of the IASI observations. These temperatures are given on a static pressure level grid. More information on ERA5 temperatures is provided in Sect. 3. We chose ERA5 because it is the most complete
homogeneous dataset of temperatures available, allowing us to select observations in any year and any type of sky. A set of 50 000 different observations is used to assess the quality of the ANN at the end of the training.

The temperatures are computed at 11 fixed pressure levels: 2, 7, 10, 20, 30, 70, 100, 200, 400, 550 and 750 hPa. These were chosen based on the Jacobians of the 231 selected channels, shown in Figure 2. The Jacobians show the sensitivity of IASI
channels to the temperature profile in K/K. We see in Figure 2 that the Jacobians peak at different altitudes, in particular for pressures <10 hPa suggesting that IASI has a good information content at these altitudes. The pressure levels were chosen based on a tradeoff between having equally distributed levels along the vertical while matching the maxima of the Jacobians of the selected channels. The pressure levels are shown in dotted horizontal lines in Figure 2. We tried different



configurations for the ANN, changing the number of epochs for the training and the number of neurons in the two hidden layers. The configuration giving the best results is 5000 epochs and 150 neurons in the two hidden layers. As a result, we have an ANN with 233 neurons in the input layer (231 radiance values, 1 scan angle and 1 $CO_2$ value), 150 neurons in each of the two hidden layers, and 11 neurons in the output layer (the 11 pressure levels shown in horizontal lines in Figure 2).

We used the trained ANN to retrieve temperatures from all IASI observations between 2008 and 2020. After the temperature profiles are computed, we use a static filter based on ERA5 mean surface pressure between 2008 and 2020 to account for orography (as some high altitude regions do not have temperature at 750 hPa for instance).

## 3 Results

Atmospheric temperatures between 2008 and 2020 were computed at the 11 pressure levels. We use Metop-A observations until 2017 and Metop-B observations for 2018 onwards. Metop-A satellite is now exploited on a drifting orbit since June 165 2017, in order to extend its lifetime to 2022 (EUMETSAT, 2018). We compare the whole IASI time series with the temperatures from ERA5 reanalysis, from the Analyzed RadioSoundings Archive (ARSA) and from EUMETSAT.

### 3.1 Comparison with ERA5

The European Center for Medium Weather Forecast (ECMWF) reanalysis, ERA5 (Hersbach et al. 2018, Copernicus Climate Change Services) is a 4D-Var data assimilation product. It is part of the Integrated Forecast System (IFS) that provides 170 variables relevant to the atmosphere, land and ocean. ERA5 atmospheric temperature product used in this work is hourly, and is available on 37 pressure levels (from the surface up to 0.01 hPa). ERA5 actually assimilates IASI radiances from Metop-A and Metop-B, as well as high spectral resolution radiances from other instruments such as AIRS on Aqua, and CrIs from S-NPP and NOAA-20. We note that IASI is the largest contributor to error reduction for global numerical weather prediction in the thermal infrared spectral band (Borman et al., 2016).


ERA5 temperatures are given on a 0.25°×0.25° latitude-longitude grid. For the comparison with the IASI ANN output, ERA5 temperatures were interpolated to the time, latitudes and longitudes of IASI observations. We then computed the daily zonal mean of the IASI-based ANN temperatures and ERA5 and looked at the differences between the two datasets. Figure 3 illustrates the zonal mean differences between the ANN retrievals and ERA5 from 2008 to 2020 for the 11 pressure levels 180 considered in this study.

Between 200 hPa and 750 hPa, the differences are small at all latitudes (less than 0.5 K). We see slight seasonal variations of the differences at 200, 400 and 550 hPa. At 750 hPa, the seasonal variations are more pronounced and more often negative than at the other pressure levels. Note that due to orography at this pressure level, there is no data over the South Pole,

Greenland and in the major mountain ranges.

Between 7 hPa and 100 hPa, the differences between the two datasets are much smaller (less than 0.5 K) at mid latitudes and at the poles. Around the equator (30°S to 30°N), the differences are slightly larger (1 K). The larger differences over the equator can be explained by the high cloud cover in this region, as high water vapour concentrations contaminate the
radiances of the whole IASI spectrum (Parracho et al., 2021).

At 2 hPa, the differences are about 2K (variation of positive and negative differences). This is because the IASI channel selection is less sensitive to temperature changes at these pressure levels. In Figure 2, we see that the Jacobians peaking at these pressure levels are large, but there are few of them, compared to other pressure levels, where the Jacobians are smaller,
but more numerous.

Averaged over the whole time series and at all latitude, the mean differences between the ANN output and ERA5 are -0.06,  0.11,  0.00, -0.02, -0.01, -0.01,  0.03,  0.07, 0.04, 0.01 and -0.09 K for each pressure level from 2 to 750 hPa, suggesting a small total bias between the two datasets.

Since averaging the differences over longitudes makes them smaller, we looked at the daily spatial differences. We gridded the ANN retrievals and ERA5 (interpolated to IASI coordinates) on a 1°×1° latitude-longitude grid and computed the Root Mean Square (RMS) of the daily differences in each of the 1°×1° pixel of the grid and at each pressure level. Figure 4 shows the RMS of the daily differences for the 2008-2020 period.

At 750 hPa, the RMS are small at the equator (about 0.5 K) and larger at higher latitude (between 1 and  2 K), especially around mountain ranges, where they reach 3 K. Between 550 hPa and 200 hPa, the RMS are small almost everywhere. There are regions (in particular the Antarctica, Greenland and the Himlaya at 550 hPa) where the RMS are larger and they can reach 2 K. Between 100 hPa and 7 hPa, the RMS are small at high latitude (0.5 K) and large at the equator (between 1.5 and
2 K). At 7 and 10 hPa, the band at the equator with larger RMS reaches higher latitudes (about 50°N and S). The large RMS correspond to the high differences seen at the equator in Figure 3. At 2 hPa, the RMS are between 2 and 3 K everywhere, which is coherent with Figure 3.

## 3.2 Comparison with ARSA

The Analysed RadioSoundings Archive (ARSA) is a 41-year (1979-2019) database of radiosonde temperature profiles
measurements from different stations around the globe (Scott et al., 2015). The raw radiosonde observations go through severe multistep quality controls, to eliminate gross errors. Whenever and wherever the selected radiosonde measurement is unable to provide forward radiative transfer modelers with the required information (above 300 hPa for water vapour and



above 30 hPa for temperature), ARSA combines existing radiosonde measurements with other reliable data sources in order to complete the description of the atmospheric state as high 0.0026 hPa. Temperature and water vapour profiles are thus
extrapolated with ERA-Interim outputs between 30 hPa and 0.1 hPa for temperature and between 300 hPa and 0.1 hPa for water vapour. Above 0.1 hPa, these same profiles are extrapolated up to 0.0026 hPa using a climatology of ACE/Scisat Level 2 temperature products. ARSA was validated against IASI observations by simulating spectra from the 4A/OP forward model (Scott and Chédin, 1981) with ARSA profiles as inputs, and comparing them with space-time colocated IASI observations. The pertinence of the requested modifications after this validation has been also assessed against the TIROS
Operational Vertical Sounder (TOVS), the Advance TIROS Operational Vertical Sounder (ATOVS, Reale et al., 2008), the Atmospheric InfraRed Sounder (AIRS, Lambrigtsen et al., 2004), the High resolution Infrared Radiation Sounder (HIRS4, EUMETSAT, 2013b), and the Microwave Humidity Sounder (MHS, Hans et al., 2020). Based on these validations, incorrect or unreliable data inherent to the quality of the radiosondes (e.g. water vapour above 300 hPa) or temperature data above 1 hPa were completed with nominal or revisited ECMWF ERA-Interim reanalysis (Dee et al., 2011) data (water vapour
profiles) or other relevant auxiliary datasets (in particular Level 2 results of ACE-FTS temperature profiles above 10 hPa) measurement data. ARSA provides a 43 pressure-level profile (from the surface to 0.0026 hPa) of temperature, water vapour and ozone, and surface temperature. It is useful to recall that ARSA is being reprocessed to substitute ERA5 for ERA-Interim. This will allow, among other things, to extend the period beyond summer 2019, when the production of ERA-Interim stopped.


For the comparison with our IASI retrievals, we interpolated ARSA temperature profiles to the 11 pressure levels of the ANN and we only kept the stations for which there were at least 300 observations per year between 2008 and 2018. Figure 5 shows the positions of these stations.

We then added the 14 stations present in Antarctica and the 6 stations in Greenland to have observations at high latitudes, although these stations have less than 300 observations per year. The stations in Greenland have between 150 and 300 observations per year on average so the time coverage is still satisfactory. However, in Antarctica, the stations have between 10 and 150 observations per year (only two stations have more than 100 observations per year), so the time coverage is very low.


We divided the stations into 8 distinct regions, and we computed the daily mean temperature of all the observations of each region. We interpolated IASI temperatures to the latitude, longitude and time of each considered station, and we computed the daily mean IASI temperature in each region. We then computed the differences between the two datasets.

Figure 6 shows the daily differences between IASI retrievals and ARSA mean regional temperature in the 8 selected regions between 2008 and 2018. At 2 hPa, we see large positive differences of more than 2 K in all the regions, as for ERA5.

Between 7 and 100 hPa, the differences are small and mostly negative (about 0.5 K). At 200 hPa and below, the differences remain small and negative in the Pacific, Oceania and East Asia. In Greenland, North America and Europe, the differences at these pressure levels are slightly larger and more often positive (about 0.5 K, up to 1 K in North America and Europe) than

in the other regions.

In Antarctica and, to a lesser extent, the Arabian Peninsula, there are more daily variations of positive and negative differences, and they are a little larger (about 0.7 K in the Arabian Peninsula and 1 K in Antarctica) than in the other regions. This can be because of the low space (few stations) and time coverage (only for Antarctica) in these regions. However, we

see the same pattern than in the other regions: large differences at 2 hPa, small differences at 7 hPa and lower, more positive differences in the troposphere.

Figures 3, 4 and 6 show that between 7 and 750 hPa, the IASI ANN product gives good quality temperatures, very consistent with the temperatures of the ERA5 and ARSA datasets (differences inferior to 1 K at most latitudes, 2 K at the equator). At 2

hPa, the quality of the ANN product decreases, as it was reflected in the lower count of Jacobians of IASI (Figure 2). This means that at 2hPa, the temperatures are not accurate enough to follow the long-term evolution of atmospheric temperatures. However, they can still be used to study large variations of temperature (during extreme events for example).

Figure S1 shows the differences between ERA5 and ARSA over the same period and in the same regions, with ERA5

interpolated to the latitudes, longitudes and time of the ARSA observations. The differences between ERA5 and ARSA are very similar to those between IASI retrievals and ARSA, but slightly smaller (less than 0.3 K between 750 and 7 hPa, 2 K or more at 2 hPa).

### 3.2 Comparison with the EUMETSAT reprocessed temperature record

In 2020, EUMETSAT reprocessed the IASI temperature record (doi:10.15770/EUM_SEC_CLM_0027), so it is now

homogeneous over the whole IASI time series (EUMETSAT, 2020). The reprocessed temperatures were computed with a Piece-Wise Linear Regression Cube (PWLR$^3$) algorithm, using all IASI observations in input (clear and cloudy scenes), and observations from two other microwave instruments flying onboard the Metop-A and -B satellites: the Microwave Humidity Sounding (MHS) and the Advanced Microwave Sounding Unit (AMSU-A).

The basic principle of this algorithm is a linear regression between IASI radiance observations and real atmospheric temperatures. To take into account the non-linearity between the observations and the temperatures, the training dataset is divided into several several sub-datasets, resulting from a k-mean clustering. This ensure that, in each sub-dataset, a linear relationship is a good approximation between the observations and the temperature and different linear regression coefficient are computed for each sub-dataset.




We compared the ANN retrievals with this reprocessed EUMETSAT dataset. Since the two methods use the same IASI observations input, there is no need for an interpolation over the coordinates of the observations. However, EUMETSAT temperatures profiles are retrieved on 138 levels, reflecting the 137 hybrid levels from the ERA-5 L137 grid plus the surface level, so we interpolated EUMETSAT temperatures to the fixed pressure levels of the ANN. Figure 7 shows the differences

between the zonal mean temperatures of the ANN output and EUMETSAT.

The differences are small at all pressure levels (less than 0.5 K) except at 2 hPa where they can reach 1 K, and we see seasonal variations of the differences that are more pronounced in the troposphere (750, 550 and 400 hPa). At 7, 10 and 70 hPa, the differences are positive and larger at the equator and they decrease over time. This bias can also be seen at higher

latitudes and other pressure levels, although it is less obvious. This bias might be due to the fact that EUMETSAT's algorithm does not use $CO_2$ in input and the retrieval is impacted by the variations of the $CO_2$ over time, which we account for.

Note that although the ANN and EUMETSAT retrievals are both based on IASI radiances, the two temperature records are

not redundant: EUMETSAT will keep on doing minor updates but not reprocessed the data back in time, whereas our dataset can constantly be enhanced and updated rapidly. Moreover the two datasets use different observations (IASI radiances and $CO_2$ concentrations for the ANN, and IASI, AMSU and MHS radiances for EUMETSAT) and different methods of retrieval (ANN and PWLR[3]).

## 4 Applications

We used the ANN temperatures to compute trends over the past 13 years. One of IASI's main asset is its high radiometric stability over the years and it is used as a reference for the inter-calibration of infrared sensors by the Global Space-Based Inter-Calibration System (Golberg et al., 2011) so temperatures derived from IASI radiances are a good product to study atmospheric trends.

We use IASI daily zonal mean temperature (latitude bands of 1°) and we compute the Theil-Sen estimator for each latitude and each pressure level. The Theil-Sen estimator is a robust method for computing linear trends, where the trends is determined by the median of all the possible slopes between pairs of points (Theil, 1950; Sen, 1968). We also computed the associated p-values, with a 0.05 threshold for significance being considered. Figure 8 shows the significant temperature trends for the 2008-2020 period. Non-significant trends are shown in grey in Figure 8.


We clearly see a warming in the troposphere. At the equator, temperature increases by 0.7 K/decade in the upper





troposphere, and 0.5 K/decade in the lower troposphere. At mid latitude, the warming is weaker. As highlighted by previous studies (IPCC, 2014), the poles are where tropospheric temperatures are warming the quickest, especially the Arctic, where temperatures increase by 1 K/decade (arctic amplification). The values of the trends we found between 45°S and 45°N are

similar to those found by Shangguan et al. (2019), although the areas of strongest warming are slightly different.

In the stratosphere, we observe a cooling almost everywhere except from 40°S northwards. The cooling is strongest at the equator, above 20 hPa (-1 K/decade). In the Arctic, there is no significant trend. This cooling of the stratosphere has also been observed by Maycock et al. (2018) and Randel et al. (2016), although their values are inferior to those found here (but

their time period is also different). In the Southern Hemisphere, we see two areas of warming: a strong one at 50°S and 100 hPa (1 K/decade) and another located at 80°S and 10 hPa (0.4 K/decade). Part of this warming is due to a sudden stratospheric warming (SSW) that happened in September 2019 (Safieddine et al., 2020b). The temperature increased by more than 30 K in a few days. Such a warming toward the end of our study period has a strong impact on trends. However, the computation of trends over the period 2008-2018 still shows warming in these two regions and it cannot be attributed to

the SSW. The warming is weaker at 50°S-100 hPa (0.6 K/decade) and stronger at 80°S-10 hPa (0.8 K/decade). Due to the Montreal Protocole in 1987, the ozone hole has been recovering since the 1990's (WMO, 2018; Weber et al., 2018; Strahan et al., 2019), and warming in the stratospheric south pole can be attributed to this recovery.

We also computed trends with ERA5 and ARSA (supplementary material, Figure S2 and S3). ERA5 trends are very similar

to those of IASI retrievals, except for a strong warming over the Arctic between 2 and 7 hPa (1 K/decade). With ARSA, we see a warming between 0.5 and 1 K/decade in Antarctica at all altitudes except 2 hPa. In Greenland, we also see a warming at almost all altitudes, but weaker (between 0 and 0.5 K/decade). In all the other regions, we see a warming between 750 and 200 or 100 hPa and a cooling above 100 hPa. The tropospheric warming and stratospheric cooling are more important in the Pacific than in the other regions. In the Arabian Peninsula, we see a smaller warming below 200 hPa than in the other

regions. These results are coherent with the trends computed with IASI and ERA5.

## 5 Conclusion

We use an artificial neural network to construct a homogeneous temperature record from IASI radiances. This dataset is available from https://iasi-ft.eu/products/atmospheric-temperature-profiles/ (doi for Metop-A temperatures:10.21413/IASI-FT_METOPA_ATP_L3_LATMOS-ULB    and    doi    for    Metop-B    temperatures:

10.21413/IASI-FT_METOPB_ATP_L3_LATMOS-ULB). Validation of the IASI ANN product with ERA5, ARSA and EUMETSAT reprocessed temperatures shows a good agreement between the four datasets especially between 7 and 750 hPa. The differences between IASI ANN temperatures and ERA5 are inferior to 0.5 K at most latitudes and most pressure levels, and the differences between IASI and ARSA are similar, demonstrating, if needed, that our IASI product can be used





to assess local variation of temperature and to compute trends.


We used these temperatures to compute trends over the 2008-2020 period. We found an increase of tropospheric temperatures, stronger in the equatorial upper troposphere (0.7 K/decade) and at the poles (1 K/decade due to arctic amplification). We also see a strong stratospheric cooling between 30°S and 30°N. In the Southern stratosphere (40°S to 90°S), there are two regions with important warming due to the ozone hole recovery.


This work shows that artificial neural networks are efficient to retrieve atmospheric temperatures from huge datasets of radiance. With this method, temperature profiles from all 10 billion observations from Metop-A and B can be computed in two days. With the short computation time and with IASI radiances being available a few hours after the observations, we can obtain temperature profiles in near real time.


We now have a homogeneous product to study seasonal and climatological variations of temperatures. It can also be used to study extreme events such as El Niño-Southern Oscillation, volcanic eruptions, heatwaves, sudden stratospheric warmings, and their link with climate change.

Although the trends computed with the ANN retrievals are coherent with other studies, a 13-year period is slightly too short for them to be fully reliable, and they can be impacted by short-term variation of temperatures (El-Niño Southern Oscillation for example). Chédin et al. (2016) showed that the number of years required to meet the probability assigned criterion with the Theil-Sen estimator is 14-15 years. However these results are promising: since IASI is planned to fly for at least another few years, the trends will become more and more reliable as the record gets longer. From 2024 onwards the IASI-NG
missions onboard Metop-SG (Crevoisier et al, 2014) will continue the IASI record, allowing to derive trends on longer timescales.

**Data availability**

EUMETSAT reprocessed L1C and L2 data are available at https://doi.org/10.15770/EUM_SEC_CLM_0014 https://doi.org/10.15770/EUM_SEC_CLM_0027, respectively. ERA5 data can be downloaded from the Copernicus Climate
Change Service Climate Data Store : https://cds.climate.copernicus.eu/cdsapp#!/dataset/reanalysis-era5-pressure-levels?tab=overview. Requests for ARSA temperatures can be made at : https://ara.lmd.polytechnique.fr/index.php?page=arsa. The temperatures retrieved with the ANN can be downloaded at https://iasi-ft.eu/products/atmospheric-temperature-profiles/.





## Author contributions

M.B. designed the ANN to retrieve the temperatures, performed the validation and prepared the manuscript with contributions from all co-authors. S.W and L.C. provided the reader for L1C data. F.A and V.P provided the selection S1 and O.L provided the selection S2. N.S. provided ARSA temperatures and helpful explanations on their construction. M.D.-B. provided informations on EUMETSAT L2 data. This work was supervised by S.S. and C.C.

## Competing interests

The authors declare that they have no conflict of interest.

## Acknowledgments

IASI has been developed and built under the responsibility of the Centre National d'Etudes spatiales (CNES, France). It is on board the Metop satellites as part of the EUMETSAT Polar System. The IASI L1C data are received through the EUMETCast near-real-time data distribution service. This project has received funding from the European Research Council (ERC) under the European Union's Horizon 2020 research and innovation program (grant agreement No 742909). It was also supported by the Prodex arrangement IASI.FLOW (Belspo-ESA). L. Clarisse is a research associate (Chercheur Qualifié) supported by the Belgian F.R.S.-FNRS. The LATMOS team is grateful to CNES for scientific collaboration and financial support. We thank EUMETSAT for providing us a full reprocessed dataset of radiances with the latest version of the L1C. We also thank Juliette Hadji-Lazaro and Philippe Keckhut for their help.

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

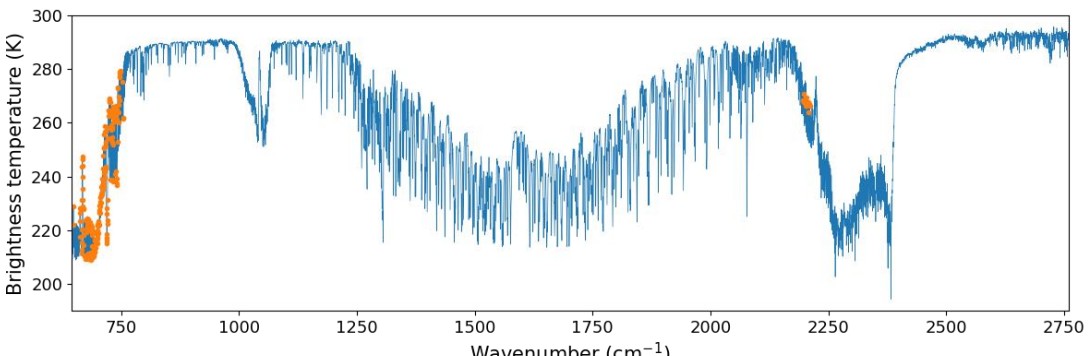

**Figure 1: IASI clear-sky spectrum in brightness temperature in K (blue), with the selected channels in orange**



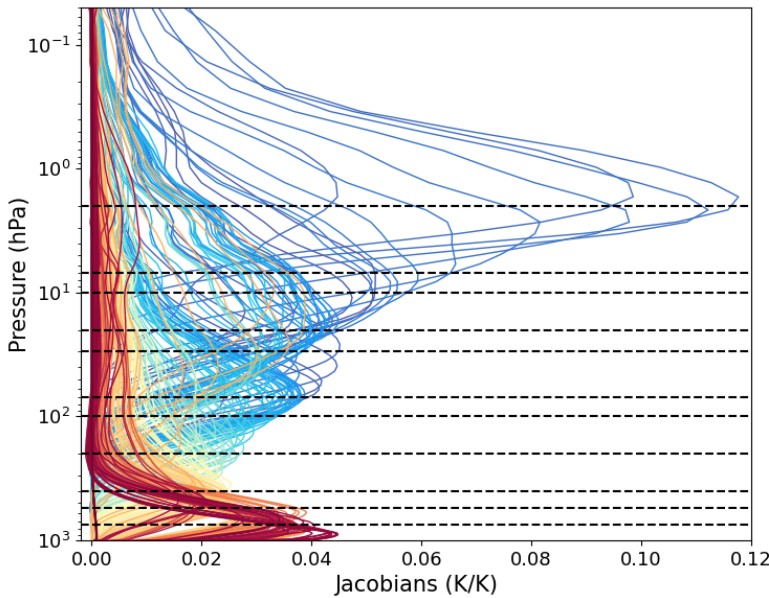


**Figure 2: Jacobians for the 231 selected IASI channels. Horizontal dashed lines represent the 11 pressure levels for which the temperatures are computed. The colors of the jacobians represent the wavenumbers of the selected channels (blue for the channels around 700 cm⁻¹, red for the channels around 2200 cm⁻¹).**







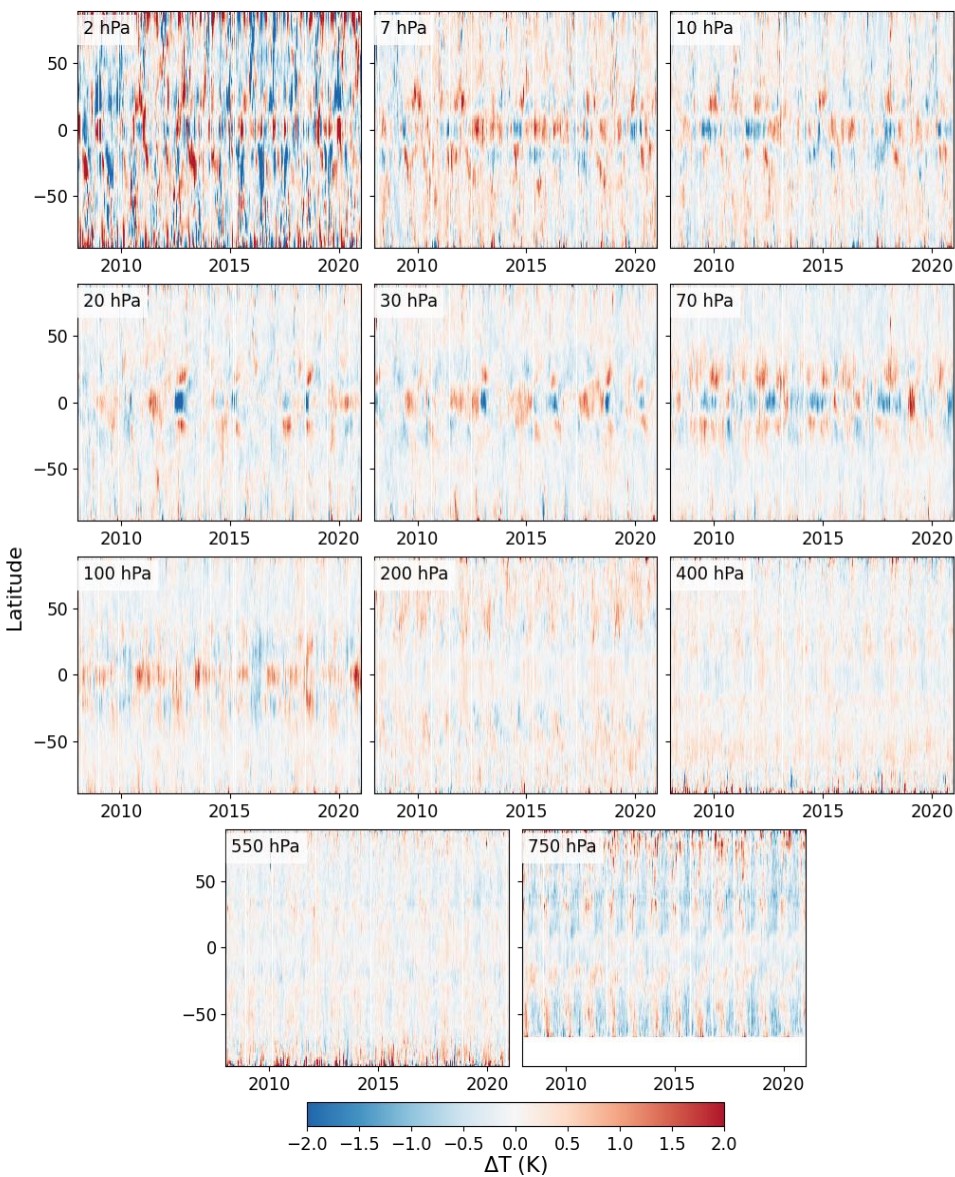

Figure 3: Daily zonal mean differences between IASI and ERA5 zonal mean temperature for the 11 pressure levels of the ANN.




**Figure 4: RMS of the daily differences between IASI retrievals and ERA5 in 1°×1° latitude-longitude grid over the period 2008-2020.**





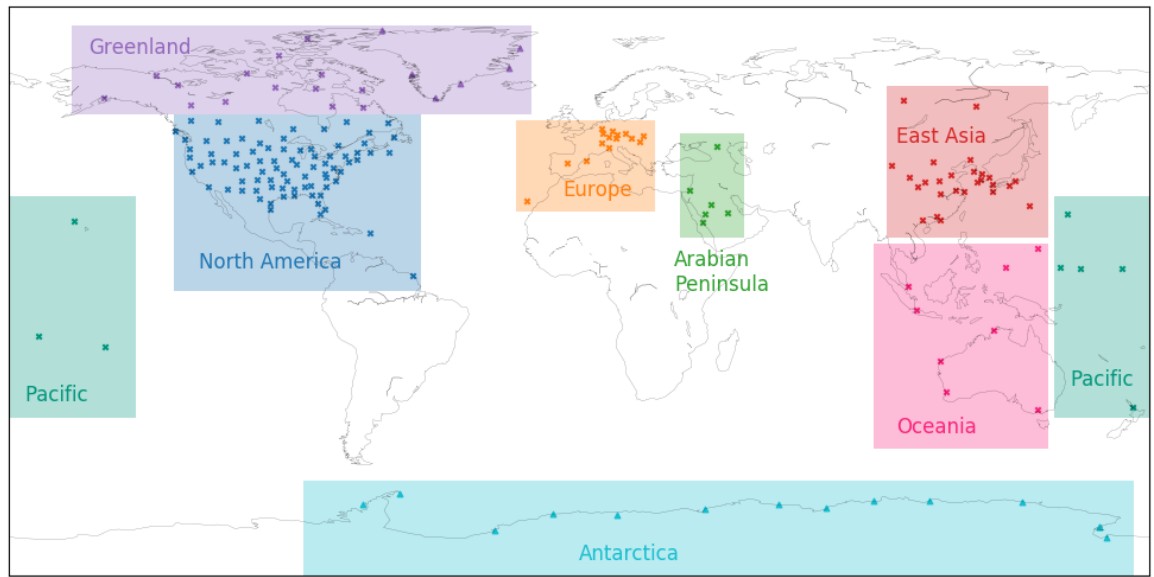

**Figure 5: Locations of the ARSA stations with at least 300 observations per year (crosses) and the stations in two other regions (Greenland and Antarctica) that do not satisfy this condition are marked in triangles. The rectangles of color correspond to the regions in which we compared IASI temperatures to ARSA.**




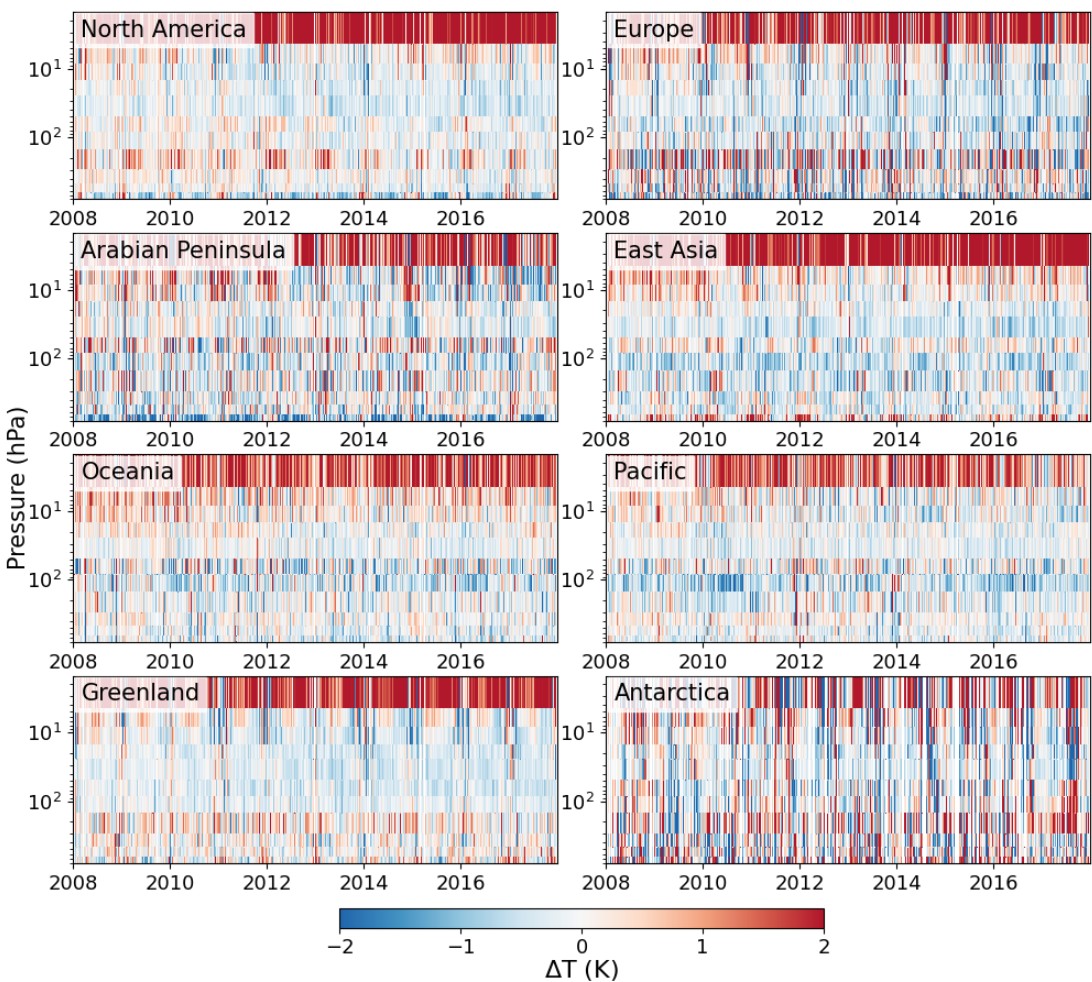

650

**Figure 6: Dayly differences between IASI and ARSA temperatures between 2008 and 2018 in North America, Europe, the Arabian Peninsula, East Asia, Oceania, the Pacific, Greenland and Antarctica.**

655





**Figure 7: Daily zonal mean differences between IASI ANN output and IASI EUMETSAT zonal mean temperature for the 11 pressure levels of the ANN.**



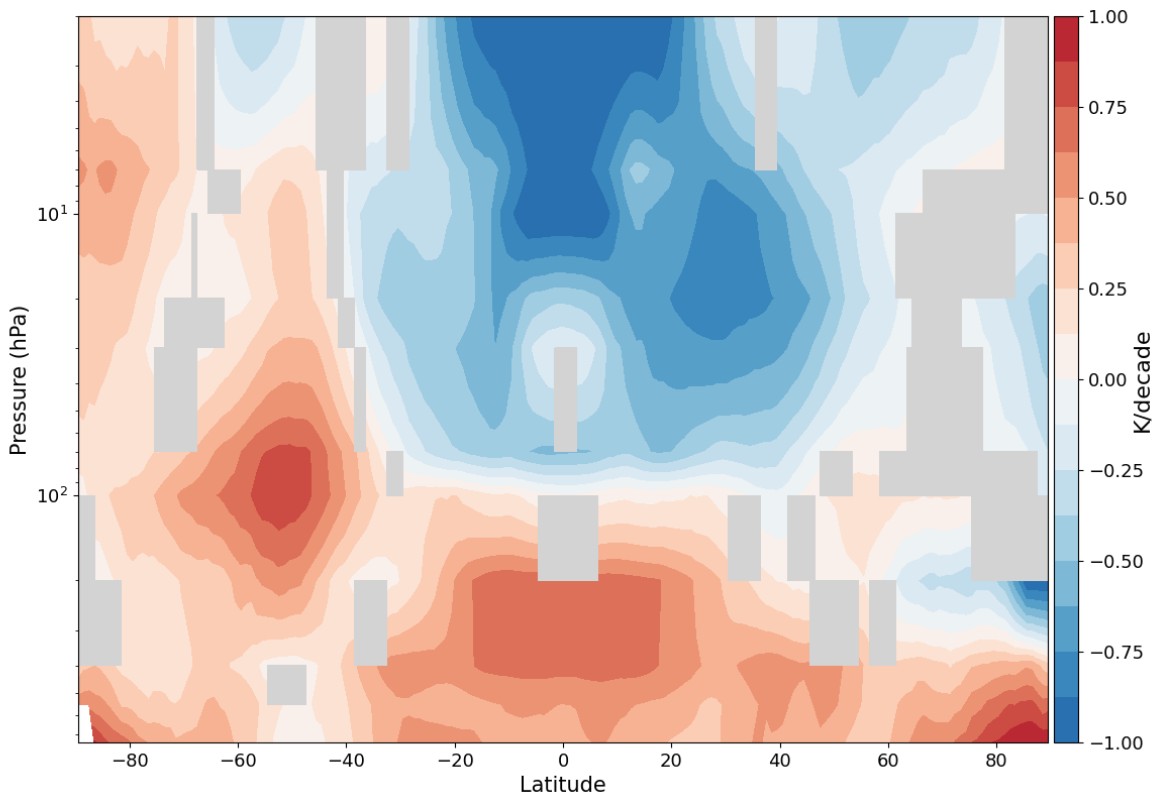

*Figure 8: Zonal temperature trends for the period 2008-2020 computed with the outputs of the ANN. Grey areas correspond to trends that are not statistically significant.*