# Peer review of "Time evolution of temperature profiles retrieved from 13 years of IASI data using an artificial neural network"

_Atmospheric Measurement Techniques, 2021_

## Author Comment (AC1)

This paper presents analysis of 13 years of temperature data from homogenized IASI measurements, based on a neural network retrieval. Results include comparisons with several other temperature data sets (ERA5, radiosondes and independent EUMETSAT retrievals), and calculation of temperature trends over 2008-2020. The new IASI retrievals show reasonable agreement with the other data sets, and trends from the short data record show the expected structure of tropospheric warming and stratospheric cooling, although with interesting detailed structure. Overall the IASI neural network retrieval seems reasonable and can provide accurate information on tropospheric and stratospheric temperatures. The paper is a useful contribution regarding the details of the data and retrieval, and it is appropriate for AMT. I have a number of comments for the authors to consider in revision.

We thank the reviewer for their useful comments. We answer each of the questions raised in blue hereafter.

The comparisons with ERA5 in Fig. 3 show systematic patterns in the tropics over levels 100-7 hPa that are suggestive of the quasi-biennial oscillation (QBO), and not tropospheric cloud cover as suggested near line 188. I would guess that IASI data have relatively low vertical resolution and underestimate some of the QBO temperature signal in ERA5. A simple comparison of IASI vs. ERA5 in height-time cross sections at the equator would clarify this behavior (showing deseasonalized temperature anomalies in both data sets, along with their differences). This may also explain the large equatorial rms differences seen in Fig. 4.

Thank you for pointing this out, you are indeed right. A figure showing the monthly differences averaged over 10°S-10°N and the corresponding zonal wind from ERA5 was added. It shows, as you suggested, that the differences between 100 and 7 hPa are correlated to the QBO, and that the neural network overestimates temperatures during the easterly phase of the QBO.

[Figure]

*Figure 4: Monthly mean of the temperature differences between IASI-ANN and ERA5 (top) and monthly zonal wind from ERA5 (bottom) between 10°S and 10°N.*

The following sentences were then added in Section 4.1:

*"Figure 4 shows the monthly mean of the differences between 10°S and 10°N from 100 to 7 hPa, as well as the monthly zonal wind from ERA5 in the same latitude range. Positive differences (ANN temperatures larger than ERA5) are correlated to negative zonal wind and negative differences are correlated to positive zonal wind.*

*This suggests that the neural network overestimates temperatures during the easterly phase of the Quasi-Biennal Oscillation (QBO)."*

I think the comparisons with radiosondes (ARSA data) should be limited to pressures 750-30 hPa, where the radiosonde measurements occur. Higher levels are simply other data sets. I don't find the complicated/noisy differences in Fig. 6 to be quantitatively very useful. Additional or complementary calculations could show the mean and rms differences for each region as a function of pressure (750-30 hPa), and perhaps include correlations to quantify the agreement between IASI and radiosondes.

This is a good remak, thank you. Figure 7 (Figure 6 in the first version) was modified and is now only showing the differences from 750 to 30 hPa, as well as the time averaged differences (as suggested by the second reviewer), and the paragraphs describing this figure were modified as follows :

[Figure]

*Figure 7: Daily differences between IASI and ARSA temperatures between 2008 and 2018 in North America, Europe, the Arabian Peninsula, East Asia, Oceania, the Pacific, Greenland and Antarctica, with the time average differences profiles on the right of each subplot.*

*"Figure 7 shows the daily differences between IASI retrievals and ARSA mean regional temperature in the 8 selected regions between 2008 and 2018, and the time averaged difference profiles. We only show differences between 750 and 30 hPa as ARSA data above 30 hPa does not always come from radiosounding measurement but from the extrapolation datasets. Between 7 and 100 hPa, the differences are small and mostly negative (about 0.5 K). At 200 hPa and below, the differences remain small and negative in the Pacific, Oceania and*

*East Asia. In Greenland, North America and Europe, the differences at these pressure levels are slightly larger and more often positive (about 0.5 K, up to 1 K in North America and Europe) than in the other regions.*

*In Antarctica and, to a lesser extent, the Arabian Peninsula, there are more daily variations of positive and negative differences, and they are a little larger (about 0.7 K in the Arabian Peninsula and 1 K in Antarctica) than in the other regions. This can be because of the low space (few stations) and time coverage (only for Antarctica) in these regions. However, we see the same pattern than in the other regions: large differences at 2 hPa, small differences at 7 hPa and lower, more positive differences in the troposphere.*

*In all the regions, the time averaged differences range from -0.6 K to 0.6 K except in the Arabian Peninsula at 750 hPa where they reach 0.9 K."*

A new figure (Figure 8) showing the standard deviation of the differences and the correlation between the two datasets was added. The following discussion was added to the text:

*"Figure 8 show the standard deviation of the daily differences between IASI-ANN and ARSA temperatures and the correlation between the two datasets in the 8 regions. In most regions, the standard deviation ranges from 0.5 to 1 K, except in the Arabian Peninsula and East Asia where they reach 1.3 K at 750 hPa, and in Antarctica and Europe where they range from 1 to 2 K. The correlations between IASI-ANN and ARSA temperatures show that there is no significant bias between the two datasets."*

[Figure]

*Figure 8: Standard deviation profiles of the differences between IASI-ANN and ARSA temperatures (left of each subplot) and correlation between the two dataset (right of each subplot).*

Several comments on the trend results in Fig. 8:

1- Results are shown for simple linear trends, but it would be good to test the sensitivity to including additional terms in the regression that explain known variations in tropospheric and stratospheric temperatures, including the QBO and ENSO. This is standard practice in calculation of long-term trends, e.g. Steiner et al 2020 (DOI:10.1175/JCLI-D-19-0998.1). The tropical stratospheric trends look to me to have structure possibly aliased from the QBO in this short record.

Figure 10 (Figure 8 in the previous version of the manuscript) shows trends computed without the contribution of ENSO and QBO, that were removed with a multiple linear regression. The main change is the reduction of the warming trend in the equatorial troposphere, that was mostly due to the El-Nino event of 2015-2016. However, stratospheric trends do not seem to be significantly impacted by the QBO.

[Figure]

*Figure 10: Zonal temperature trends for the period 2008-2020 computed with the outputs of the ANN. Grey areas correspond to trends that are not statistically significant. The dotted rectangles represent the regions for which the time series are shown in Figure 11.*

The main change from the figure in the previous version is a weaker warming in the equatorial troposphere. The text was modified accordingly:

*"We use IASI daily zonal mean temperature (latitude bands of 1°) and we compute the Theil-Sen estimator for each latitude and each pressure level. The Theil-Sen estimator is a robust method for computing linear trends, where the trend is determined by the median of all the possible slopes between pairs of points (Theil, 1950; Sen, 1968). We also computed the associated p-values, with a 0.05 threshold for significance being considered. Before the Theil-Sen estimator was applied, we removed the contributions of El Niño-Southern Oscillation (ENSO) and the QBO to temperatures. Their contribution was computed using a multiple linear regression based on the Multivariate ENSO Index (MEI, https://psl.noaa.gov/enso/mei/) and the QBO30 and QBO50*

*indices (equatorial zonal winds at 30 and 50 hPa, https://www.cpc.ncep.noaa.gov/data/indices/). Figure 10 shows the significant temperature trends for the 2008-2020 period. Non-significant trends are shown in grey in Figure 10.*

*We clearly see a warming in the troposphere. In the tropics, we see a warming of 0.2-0.3 K/decade. At mid latitude, the warming is stronger (0.5-0.6 K/decade). As highlighted by previous studies (Masson-Delmotte et al., 2021), the poles are where tropospheric temperatures are warming the quickest, especially the Arctic, where temperatures increase by 1 K/decade (arctic amplification). The values of the trends we found between 45°S and 45°N are similar to those found by Shangguan et al. (2019)."*

2- The calculated tropospheric temperature trends are reasonably close to results shown in Steiner et al 2020 based on radiosonde and GPS RO data sets for the period 2007-2018 (their Fig. 11). However, these trends are much larger than corresponding results for longer time series, and are certainly influenced by the short data record and large warm ENSO event occurring in 2016, near the end of the time series. This detail should be clarified, as trends in excess of 0.7 K/decade are not representative of long-term tropospheric trends. In addition to Fig. 8, it could be helpful to include time series at several specific locations (tropical upper troposphere, Arctic troposphere, SH lower stratosphere – see below) to show the actual behavior and provide perspective to the trends calculated from this short record.

Figure 11 shows the time series in the 4 rectangles of Figure 10. The strong warming observed in the equatorial troposphere was indeed driven by the 2015 El-Nino, and the warming is significantly reduced when ENSO contribution is removed. In the other regions, ENSO and QBO do not seem to have a significant impact on the evolution of temperatures.

[Figure]

*Figure 11: Times series of temperatures in the four rectangles of Figure 10 (blue) and time series without ENSO and QBO conttibutions (orange). The exact locations of the four regions are 45°S-60°S and 100-70 hPa for Stratosphere Southern mid latitudes, 80°S-90°S and 10-7 hPa for Antarctic stratosphere, 20°S-20°N and 550-200 hPa for Tropical troposphere, and 75°N-90°N and 750-550 hPa for Arctic troposphere.*

The following paragraph was added in Section 5:

*"Figure 11 shows the temperature time series in the regions delimited by dashed rectangles in Figure 10. The time series are shown with and without the contributions of ENSO and QBO. In the equatorial troposphere, we see that temperature are increasing. However, removing the contribution of ENSO significantly reduces the warming trend, as most of it was driven by the strong El Niño event of 2015-2016. In the Arctic troposphere, there is no significant differences in the time series with and without the contribution of ENSO/QBO, suggesting that these phenomena does not have a large impact on Arctic temperature, and the warming observed in this region is due to the increase of greenhouse gases and Arctic amplification. In the Southern stratosphere, the trend in both warming regions seem to be driven by the 2019 SSW. However, we see a continuous increase of temperatures before 2019 (with and without ENSO/QBO contribution) that cannot be attributed to the SSW and is most likely due to ozone hole recovery."*

3- The large warming trend in the SH lower stratosphere (50 S, 100 hPa) is probably a result of transient warming in early 2000 (2020?) tied to the Australian New Year fires (Yu et al, 2021, doi: 10.1029/2021GL092609). This could easily be confirmed by examining the associated time series.

Figure 11 shows that the warming observed at 50°S/10hPa happened before 2020. Some of the warming happens continuously from 2008 to 2019 and is due to ozone hole recovery, and the 2019 SSW also had an impact on the observed warming trend. There could be an increase of temperatures observed in early 2020 (before a decrease in the second half of 2020) but it is complicated to distinguish the contribution of the Australian fires from the general warming trend.

4- I suggest adding a line in Fig. 8 indicating the time average tropopause.

A line indicating the time average tropopause height between 2008-2020 (from MERRA2 reanalysis) was added on Figure 10:

[Figure]

*Figure 10: Zonal temperature trends for the period 2008-2020 computed with the outputs of the ANN. Grey areas correspond to trends that are not statistically significant. The dotted rectangles represent the regions for which the time series are shown in Figure 11.*

---

## Author Comment (AC2)

In this paper, the authors describe the creation of a global temperature dataset derived from IASI satellite observations. A new retrieval method based on using an Artficial Neutral Network (ANN) to retrieve vertical profiles of atmospheric temperature is applied globally for 13 years of IASI data from 2006-2018 at 11 pressure levels throughout the troposphere and stratosphere. They then verify their derived temperatures against ERA5 reanalyses globally and ARSA radiosonde observations in selected locations. They find that their new retrieval typically deviates less than 0.5 - 1K from these reference data. Finally they fit linear trends to their retrieved temperatures to show global warming/cooling rates in the troposphere and stratosphere, providing further important evidence and quantification of our changing climate.

Overall, the paper and its dataset appear to be of very high quality and are potentially very useful to the scientific community. The manuscript is well written and in general there is sufficient detail to describe the method. Figures are generally clear and the analysis is sufficient to support the conclusions. The temperature dataset itself is very well presented and archived on a freely accessible website and easily downloaded from an FTP server.

I have some minor comments below that the authors may like to consider in a revised manuscript.

Below these, I have also included some typographic and stylistic suggestions that they may also like to consider. I do not require a point-by-point response to these typographic/style suggestions.

*We thank the reviewer for their positive review and useful comments. We answer each of the questions raised in blue hereafter.*

Minor Comments

The manuscript could benefit from a short "Data" section where the IASI instruments and radiances, the ERA5 reanalyses and the ARSA radiosonde data are briefly described. This would greatly improve flow in the results sections where the ATPs are compared to ERA5 and IASI, without needing to introduce and describe them in the text there.

*This is indeed a good suggestion. A data section presenting IASI radiances, ERA5 reanalysis and ARSA measurements was added as Section 2, and the paragraphs describing the data later in the article were removed:*

*"2 Data*

*2.1 IASI radiances*

[revised manuscript text omitted]

Did the authors notice and difference in the accuracy or noise levels in their retrieval during day or night conditions? Possibly, I might expect a local nighttime retrieval to be better constrained than daytime because local thermodynamic equilibrium can be assumed at night, however this might not be a problem for this retrieval because of the machine learning method applied. The authors could simply regenerate Fig. 3 once for ATPs derived during local daytime and again for local nighttime conditions, and see if there are any differences. They can just report this in the manuscript if the is/isn't any difference, no need to include the figures. There is another temperature retrieval for the NASA AIRS satellite developed by Hoffmann and Alexander (2009) who found a large difference between day/night, but theirs was a different method to the one applied here.

The two figures below show the differences between IASI-ANN and ERA5 temperatures in 2016 for day and night observations separately. There are no significant changes in the differences when looking at day or night observations. We add this in Section 4.1 as follows:

*"When taking day and night observations separately (not shown), there is no significant change of the differences."*

[Figure]

Figure: Daily zonal mean differences between IASI-ANN and ERA5 temperatures for day observations only.

[Figure]

Figure: Daily zonal mean differences between IASI-ANN and ERA5 temperatures for night observations only.

Sect. 2.2, see also l.171-174 - Is it a problem that the training dataset output (ERA5) actually assimilates IASI radiances, so it sounds like the later comparison to ERA5 could be a bit circular? I don't think this presents a problem, but the authors should briefly explain why this is not a problem.

The following sentence was added at the beginning of Section 4.1:

*"Although the neural network was trained with ERA5, it does not reproduce the same temperatures. The output of the retrieval is mainly governed by the variations of observed radiances, and ERA5 can be used for validation."*

l.144 - "More information on ERA5 temperatures..." this is where a dedicated Data section would be more helpful.

This sentence was removed, as the information about ERA5 was provided before in Section 2.

l.149 - I could be wrong about this, but I think what is plotted in Fig. 2 would be better described as the "weighting functions", "kernel functions" or "sensitivity functions" of the selected IASI channels. These channels, or rather their sensitivity functions with pressure, are arranged into a Jacobian matrix, but the channel sensitivity profiles themselves are not necessarily "Jacobians". Terms like "weighting function" are commonly used to describe these sensitivity functions for hyperspectral imagers, so I might suggest either using this, or clearly explaining why they are being described as Jacobians here. As I said, I may be wrong, but some more explanation around the chosen terminology is required to clear this up.

All the occurrences of the word "jacobian" were replaced by "weighting function".

Fig. 2 - Related to point 2 above, do these sensitivity functions change significantly under day/night conditions? If so, could this affect the true height of the retrieved temperature? I don't expect this effect to be very large, but the authors could comments on this.

The figure below shows the weighting functions averaged for 1000 random day observation (left), for 1000 random night observations (middle) and the differences between the two (right). There are no significant differences between day and night weighting functions. For one given channel, the differences between day and night range from -0.001 to 0.001 K/K while the maximum value of the weighting functions go from 0.04 to 0.11 K/K.

[Figure]

The following sentence was added:

*"The weighting functions do not change significantly under day or night conditions and this does not have an impact on the retrieval."*

Sect. 3.1, l.168-171 - This information for example would be better in a new "Data" section.

This paragraph was moved to Section 2.

l.178 and elsewhere - [Important] I cannot find anywhere in the manuscript where it is clearly stated whether these differences are "IASI minus ERA5" or "ERA5 minus IASI". As a result, it's not clear for sure which dataset has a warm/cold bias with respect to the other. This is the same for Figs. 3, 6, 7 and S1. I would suggest simply writing (IASI minus ERA5) or similar in the figure caption, that would be enough to clarify.

The figures showing the differences were modified so that the label of the colorbar now indicates "$T_{\text{IASI ANN}} - T_{\text{ERA5}}$ (K)" instead of "$\Delta T$ (K)". (see Figure 3 below for example)

l.191 - This first sentence is not clear, please rephrase. How about "At 2hPa, differences range from -2K to 2K globally", or similar?

The sentence was modified as suggested.

l.196-199 and Fig. 3 - The information in these lines and Figure 3 could be easily summarised in a line plot that could be included neatly into the bottom right hand corner of Fig. 3. The additional panel could show the time-averaged difference (x-axis) against latitude (y-axis) for each of the 11 pressure levels considered, which could be colour coded. This would be a very useful summary of the information contained in Fig. 3.

Showing the time averaged differences of the 11 pressure levels on a single plot was making the figure difficult to read and interpret. For more clarity, the time averaged differences were added on the right of each subplot.

[Figure]

*Figure 3: Daily zonal mean differences between IASI and ERA5 zonal mean temperature for the 11 pressure levels of the ANN, with the time averaged differences on the right of each subplot.*

l.201-204 - Does the fact that a latitude-longitude gridding is being used affect the RMS calculation? Obviously, a 1x1 degree lat-lon bin is much smaller at high latitudes than in the tropics, so there will be fewer IASI data points going into it. Are there sufficient numbers of points in each bin that the RMS is not likely to be affected by this?

On average, there are between 20 and 25 points per bin each day, and it can go up to 27 for latitudes around 75°N or S. At latitudes larger than 75°, the number of points per bin decreases and there are bins with less than 5 points, but these only concern latitudes larger than 88°.

l.206-207 - The increased RMS above mountain ranges could also be due to atmospheric gravity wave (GW) activity. If there are orographic GWs present over the mountains in the IASI measurements that are even slightly different from those simulated in ERA5 in terms of phase, amplitude, intermittency or location, this will likely result in a higher RMS value than a region with low GW activity.

The sentence was modified:

*"At 750 hPa, RMS values are small at the equator (about 0.5 K) and larger at higher latitude (between 1 and 2 K), especially around mountain ranges, where they reach 3 K and can be due to gravity wave activity."*

Sect 3.2 l.214-234 - Firstly, the description of the ARSA radisondes could go in a Data section. Secondly, there is a long paragraph here from l.216-231 where it is not entirely clear what processes are applied to the radisonde datasets by whom and which of these steps are relevant for the present study. l.231 onwards "ARSA provides a 43 pressure-level profile..." should be near the top of the paragraph for readability, or even in the Data section. The authors should also decide how best to describe the relevant quality controls and extrapolation steps applied to the ARSA data for readability, because at the moment it is a little confusing which parts are relevant.

Thank you. Indeed, you are correct. The description of ARSA was adapted and moved to Section 2 (see first comment).

l.233 - I think the authors mean "substitute ERA-Interim for ERA5"?

The sentence was modified: *"... ARSA is being reprocessed to replace ERA-Interim with ERA5"*.

l.250 and Fig. 6 - Normally it is good to show the highest time resolution possible, but would these figures be better simply showing the monthly averaged differences? This also would help to overcome the poor time coverage of the high latitude radisonde stations. The figures are also not high enough resolution to see individual daily differences anyway.

The poor time resolution is not a problem to overcome. On the contrary, averaging the differences over a month would probably underestimate the differences, as averaging the differences over the stations in each region already does.

Fig. 6 - As mentioned above, the information in Fig. 6 would be very well summarised by a line plot showing time-averaged temperature differences (x-axis) against altitude (y-axis) for each radisonde region. The different lines for the different regions could be neatly colour-coded like the authors have done in Fig. S3. I think this could be very clear and may be worth including.

As for the comparison with ERA5, putting all the profiles on the same plot was not very clear, so the profiles were added on the right of each subplot:

[Figure]

*Figure 7: Daily differences between IASI and ARSA temperatures between 2008 and 2018 in North America, Europe, the Arabian Peninsula, East Asia, Oceania, the Pacific, Greenland and Antarctica, with the time average differences profiles on the rigt of each subplot.*

l.264-265 - How much of this persistent positive temperature difference at 2hPa could be due to biases in the ARSA dataset, and not due the IASI retrieval? Looking at Fig. S1, there is a very similar positive bias when the ARSA radisondes are compared to ERA5. There could therefore be a small temperature bias at these altitudes in the ARSA data, perhaps due to the additional datasets that are used to extrapolate or constrain the radisonde data at these altitudes?

Yes, as suggested by the first reviewer, we removed the comparison above 30 hPa in the figure, because the bias probably comes from the extrapolation data so the comparison is not with radiosoundings anymore. This was clarified in Section 4.2:

*"We only show differences between 750 and 30 hPa as ARSA data above 30 hPa does not always come from radiosounding measurement but from the extrapolation datasets."*

l.269-272 - Related to the point above, it would be very useful to include in the supplementary material the exact same figure as Figs. 6 and S1 but for the differences between IASI and ERA5 for each region. This could help the authors to more confidently assess some of the observed temperature differences in different the regions.

Figure S2 shows the differences between IASI and ERA5 at the time and location of ARSA observation. The following sentences were added in Section 4.2:

*"Figure S2 shows the differences between IASI and ERA5 temperatures interpolated to the time and locations of ARSA observations. In most regions, the differences are less than 0.5 K. In Antarctica, in Europe (troposphere only) and in the Arabian Peninsula and Oceania (stratosphere only), the differences can reach 1 K."*

[Figure]

*Figure S2: Differences between IASI-ANN and ERA5 temperatures at the time and location of ARSA observations, averaged daily in each of the regions.*

Fig. 7 - [Important] The authors should explain, or at least discuss, the thin vertical red stripes that appear in some of the panels in Fig. 7 (such as in the 30hPa panel). They should clarify whether these are artefacts that result from their analysis or if they are physical. I did wonder if they were due to re-initialisations of the EUMETSAT retrieval, or perhaps even due to sudden stratospheric warmings. The authors should discuss.

The red stripes are an artefact of the analysis: this comparison was done using monthly files of ERA5 data (as the comparison between IASI-ANN, EUMETSAT CDR and ERA5 was done all at once), and for the last day of each month, when both the file of one month and the next were needed for the interpolation, there was a problem in concatenating the two files. Due to the large computing time, we did not redo the comparison. We clarified this in the description of the figure:

*"Figure 7: Daily differences between IASI and ARSA temperatures between 2008 and 2018 in North America, Europe, the Arabian Peninsula, East Asia, Oceania, the Pacific, Greenland and Antarctica, with*

*the time average differences profiles on the rigt of each subplot. The red stripes seen in some panels are artefacts from the analysis and the do not reflect a physical phenomenon."*

l.320 - "...although the areas of strongest warming are slightly different." It would be useful to briefly describe what these differences are if the authors are going to mention the Shangguan et al. (2019) study.

As suggested by the first reviewer, the contributions of ENSO of MEI were removed before computing the trends and tropospheric trends are now very similar to those found by Shangguan et al. The sentence is now:

*"The values of the trends we found between 45°S and 45°N are similar to those found by Shangguan et al. (2019)."*

l.342-345 - The DOIs listed do not appear to be working correctly, please check. Also, the authors could consider using the accepted short doi service for readability (https://shortdoi.org/).

The DOIs seem to be working, but for more clarity they were shortened to https://doi.org/hbxm (Metop-A) and https://doi.org/hbxn (Metop-B):

*"This dataset is available from https://iasi-ft.eu/products/atmospheric-temperature-profiles/ (doi for Metop-A temperatures: https://doi.org/hbxm and doi for Metop-B temperatures: https://doi.org/hbxn)."*

l.354 - Are both of these southern warming regions in Fig. 8 due to ozone hole recovery or just the region over the pole?

The warming over the pole is due to ozone hole recovery and the warming at 50°S is most likely due to the Sudden Stratospheric Warming that happened in September 2019, but both regions are impacted by the two phenomena in different proportions. The sentence was changed to "… there are two regions with important warming due to the ozone hole recovery and a SSW that happened in 2019."

One final general point, it might be very useful for the community if the authors could say something about the vertical resolution of their retrieval. Naturally, the retrieval is evaluated on 11 pressure levels, but if the authors were able to estimate the vertical resolution of the retrieved temperature at for each of these levels that would be very useful if other researchers wanted to investigate gravity wave observations in the dataset, in a similar to what has been done in many studies for AIRS (Hoffmann and Alexander, 2009). Further to this, are the retrieved ATPs also archived on the satellite scantrack or is only the global 1x1 degree grid available?

The vertical resolution goes from 5 to 12 km from the lower to the upper troposphere. In the stratosphere, the vertical resolution goes from 12 km in the lower stratosphere to 25 km above 7 hPa. This was added in Section 3.2.:

*"At the selected pressure levels, the vertical resolution goes from 5 to 12 km from the lower to the upper troposphere. In the stratosphere, the resolution goes from 12 km in the lower stratosphere to 25 km above 7 hPa."*

Only the 1°x1° grids are archived on the website.